# Effects of Forest Thinning on the Long-Term Runoff Changes of Coniferous Forest Plantation

**Hyunje Yang, Hyung Tae Choi \* and Honggeun Lim**

Forest Conservation Department, National Institute of Forest Science, Seoul 02455, Korea; yanghj2002@korea.kr (H.Y.); hgh3514@korea.kr (H.L.)

**\*** Correspondence: choiht@korea.kr; Tel.: +82-2-961-2643

**Abstract:** Forests and water are closely related to each other. Thus, forest management is crucial for the sustainable clean water supply. Forest thinning is one of the fundamental forest management practices, as it can change runoff by controlling the density of trees. In this study, the effect of forest thinning on long-term runoff changes was evaluated, based on the long-term rainfall-runoff data of a coniferous plantation forest catchment in Korea. From the double mass curve and Pettitt's test, a statistically significant increase in runoff rates was identified. A simple linear regression model of the double mass curve can successfully quantify the net effect of forest thinning on the runoff increase. Furthermore, it was also confirmed that forest thinning does not significantly increase the risk of flooding. About ten years after forest thinning, crown closure rates of the coniferous plantation forest reached a level similar to the pre-thinning period, and runoff rates returned to the pre-thinning level, due to forest growth. As a result of this study, a proposed direction for Korea's forest policy for water resource management is presented for the future.

**Keywords:** long-term runoff changes; forest thinning; forest management; water yield change; double mass curve; change point detection

---

## 1. Introduction

Forests have a large number of functions in terms of ecological and hydrological aspects, which can give ecosystem stability, mitigate hydrological risks, such as floods and soil erosion, and supply fresh water to living things. One of these functions, providing fresh water, is important because the demand for clean water is currently increasing as the quality of life improves and the population grows [1]. Many countries obtain significant amounts of their drinking water from forests, because forests normally supply high-quality water for people continuously throughout the year [2–4]. Over one-third of the largest cities globally still rely on forests for their drinking water, including New York, Vienna, Tokyo, etc. In particular, in the United States of America, over 180 million people still rely on forests to obtain drinking water [5,6]. Likewise, forests and water are closely related, and many countries have been working on forest management for clean water.

In the Republic of Korea, 64% of the country's land is covered by forests, and most of the upstream basins of major rivers, which are important sources for national water resources, are covered by forests. Therefore, most of Korea's water resources come from forests [7], and forest conservation and management are essential for a sustainable clean water supply in Korea. By the 1960s, large parts of forest lands in the Republic of Korea had been devastated by the Korean war, excessive illegal logging, and so on. To restore the devastated forest lands, the Korean government carried out several nationwide reforestation programs. About 2.2 million ha of forest—a third of Korea's forest lands—have been reforested artificially since the 1970s. The main planting species were coniferous species, such as *Pinus densiflora, Pinus rigida, Pinus koraiensis, Larix kaempferi,* and so on. At present, most of the coniferous

plantation forests have reached the fourth age class (about 31~40 years old), and managing these forests has become an important task of Korean forest policy. Although, there are many objectives in managing forests, considering that forests play an important role in sustainable water resource management in Korea, studies on how the management of coniferous plantation forests affect the water cycle and runoff characteristics are very important.

Forest thinning is a forest management practice, which is mainly undertaken to reduce forest density, increase the health of forests, and optimize the growth of individual trees [8]. This results in a number of hydrological, biological, pedological, and meteorological changes in forests [9]. From the hydrological aspect, huge parts of the water cycle in forests can be changed by forest thinning. For example, the total evaporation of forests can decrease [10–12] as the crown density reduction by forest thinning can result in the reduction of the interception of the canopy and consequently an increase in the rainfall input on the forest floor [13,14]. More rainfall inflow into forest soil can increase the possibility of increasing groundwater recharge and streamflow [9,15]. According to the hydrological potentials of forest thinning, many studies have been conducted to identify and evaluate the effect of forest thinning on the water cycle and the rainfall-runoff responses of forests, in order to mitigate water shortage and climate change impacts [12,16–18].

A paired catchment approach can be used to detect the changes in the water cycle by conducting treatments in only one of two paired catchments with a similar climate, soil properties, geology, terrain, land use, and so on [15,19]. This approach compares two catchments at the same time and can be free from the effects of climate change. Normally, the paired catchment approach is the predominant method for quantifying the effect of forest management on the water cycle, but the single catchment approach with long-term monitoring data, can also be useful [15]. The single catchment approach can identify the effects of treatments by analyzing the changes of long-term trends before, and after, forest treatments [20,21]. As forests, in particular, continuously change and grow, long-term-based analysis is essential in identifying the long-term effects of forest thinning. The availability of long-term monitoring data is an important prerequisite for applying this approach, and it is necessary to have sufficient long-term data to eliminate the long-term effects of climate change [15,22].

This study was conducted, in order to evaluate the long-term effect of forest thinning on the runoff characteristics of a coniferous plantation forest catchment, that requires forest management to increase water resources in Korea. For this purpose, two paired catchments with long-term hydrological monitoring data were used: One is a coniferous plantation forest catchment, and the other is a natural deciduous forest catchment. The coniferous forest was established over 40 years ago and was thinned when the trees were 20 years old. The natural deciduous forest catchment belongs to a national nature reserve and has not been artificially treated in any form. In order to check the effect of forest thinning on the runoff characteristics, a double mass curve analysis of long-term rainfall-runoff data was used. In addition, Pettitt's test was performed to verify that the changes in runoff characteristics are statistically significant. Finally, the changes in the long-term trend of runoff characteristics were identified from a double mass curve, using a simple linear regression model, and the effect of the forest thinning on water supply was evaluated.

## 2. Materials and Methods

### 2.1. Study Sites and Forest Thinning

The study sites are two catchments called the Gwangneung coniferous plantation (GCP; 37°45′48.23″N, 127°09′23.40″E) and Gwangneung natural deciduous (GND; 37°44′56.02″N, 127°08′56.14″E) forest catchments located in Pochun, Gyeonggi-do, Republic of Korea (Figure 1). Both the GCP and GND forest catchment are experimental catchments of the National Institute of Forest Science (NIFoS) under the Korea Forest Service. Although, both forest catchments are about 1 km away from each other, the soil and geological characteristics are the same: Sandy loam, and granite

gneiss, respectively (Table 1). The catchment areas are 13.6 ha for the GCP, and 22.0 ha for the GND forest catchment, respectively.

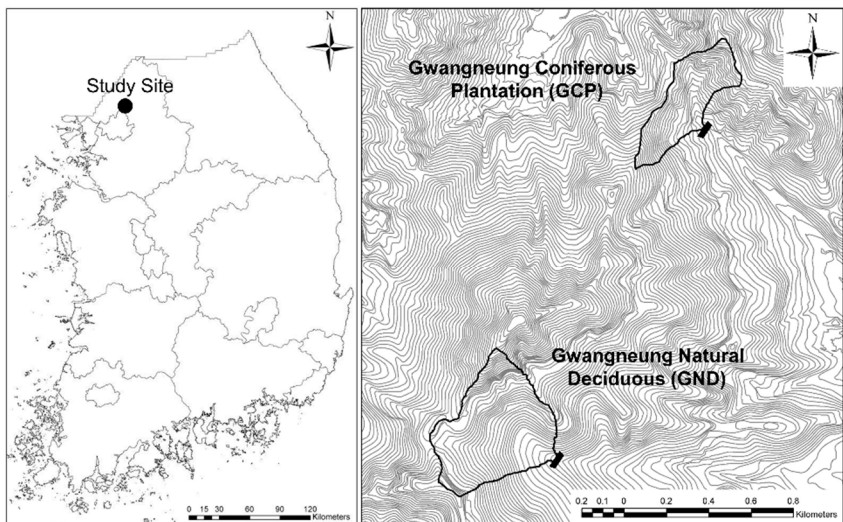

**Figure 1.** Location of the study sites: the Gwangneung coniferous plantation (GCP) and natural deciduous (GND) forest catchments.

The mean annual temperature of the GCP and the GND forest catchments is 11.2 °C; the mean annual rainfall of the GCP forest catchment is 1425 mm, and that of the GND forest catchment is 1436 mm. Both catchments lie in the temperate climate zone with four distinct seasons. The summer is humid and hot, and winter is dry and cold. Over 70% of the annual rainfall falls in the summer monsoon period between June and September, as a part of the East-Asian monsoon season.

The area of the GCP forest catchment was originally hillslope agricultural land with slash-and-burn farming. In 1976, coniferous trees, such as *Abies holophylla* and *Pinus koraiensis* were planted in most of the GCP forest catchment for reforestation. In the GND forest catchment, the predominant species are *Carpinus laxiflora* and *Quercus* spp., which are now more than 90 years old. In the GCP forest catchment, 45% of stems are uniformly thinned in all areas of the catchment. After forest thinning, the mean tree height, diameter at breast height (DBH), and stem volume slightly increased, and the growing stock volume decreased. The crown closure was 97.1% before the forest thinning, but later it decreased significantly to 70.0% (Table 2). After forest thinning, the crown closure drastically increased to 95.0% in 2003, 96.0% in 2010 and 95% in 2016.

Based on the paired catchment approach, the GCP forest catchment was selected with respect to the runoff characteristic changes from forest thinning, and the GND forest catchment was selected as a control catchment to eliminate the impacts of long-term climate change.

**Table 1.** Characteristics of the Gwangneung coniferous plantation (GCP) and natural deciduous (GND) forest catchments.

| Catchment Characteristics | GCP | GND |
|---|---|---|
| Catchment area (ha) | 13.6 | 22.0 |
| Elevation (m) | 160~290 | 280~470 |
| Bed rock | Granite gneiss | Granite Gneiss |
| Soil depth (m) | 0.3~0.6 | 0.3~0.6 |
| Predominant species | *Abies holophylla Pinus koraiensis* | *Carpinus laxiflora* Quercus spp. |
| Forest management | Planted in 1976 and Thinning in 1996 | Natural forest |
| Forest age class | V | X |

**Table 2.** Stand characteristics change of the GCP forest catchment before and after the forest thinning.

| Stand Characteristics | 1986 | 1996 | | 2003 | 2010 | 2016 |
|---|---|---|---|---|---|---|
| | | Pre | Post | | | |
| Mean tree height (m) | 6.5 | 9.6 | 10.6 | 12.2 | 13.5 | 14.8 |
| Mean DBH (cm) | 4.7 | 13.4 | 15.7 | 19.8 | 21.4 | 22.4 |
| Mean tree density (trees/ha) | 2700 | 2102 | 1147 | 1120 | 1054 | 996 |
| Growing stock volume (m$^3$/ha) | 25.2 | 150.1 | 123.8 | 205.9 | 259.1 | 283.0 |
| Mean stem volume (m$^3$/tree) | 0.009 | 0.071 | 0.108 | 0.184 | 0.246 | 0.284 |
| Crown closure (%) | 75.9 | 97.1 | 70.0 | 95.0 | 96.0 | 95.0 |

Note: The forest thinning is conducted in 1996, and Pre is pre-thinning, Post is post-thinning. DBH: diameter at breast height.

*2.2. Rainfall-Runoff Data*

Rainfall-runoff data were provided from the long-term monitoring data of the National Institute of Forest Science (NIFoS) in the Republic of Korea. NIFoS has collected rainfall-runoff data in GCP and GND forest catchments since 1981. Currently, automatic rain gauges and float-type water level recorders with a sharp-crested triangular weir (120°), in the study sites, are used for in-situ measurement. In this study, the annual rainfall and runoff data collected from 1981 to 2017 were used to analyze runoff characteristics changes. Annual rainfall and runoff values are the total amount of rainfall and water outflow throughout the year. Where data was lost for a certain period of time because of natural disasters such as typhoons and the consequent repair of the gauging station or equipment replacement, the annual rainfall-runoff data corresponding to missing data were excluded from the analysis.

*2.3. Double Mass Curve*

The double mass curve method was used to analyze the runoff characteristics that change over time. The double mass curve method is commonly used to check the consistency of hydrological data and to confirm the slope changes, as cumulative values of the rainfall and runoff data are plotted on the graph [23]. This is very practical because of the low data requirements and the simplicity of the analysis process [22]. Thus, it has been widely used to assess the effect of climate change or human interference on runoff changes [24]. In particular, when a double mass curve was plotted with cumulative rainfall and runoff data, the slope of the regression line means the runoff rate represents the runoff characteristics. Thus, from this curve, we can easily recognize the long-term runoff characteristic changes as changes in slope.

*2.4. Non-Parametric Statistical Analysis*

The double mass curve method has the advantage of making it easier for researchers to detect the change in runoff characteristics by confirming the slope change over time at a glance. However, the change point detection from the double mass curve can be so subjective, that it may lead to different results, depending on the researcher [22]. One non-parametric statistical analysis method, Pettitt's test, was performed, in order to determine the statistically objective change points [25]. This determines whether two parametric groups (population; $(x_1, \cdots, x_j)$ and $(x_{j+1}, \cdots, x_N)$) have the same tendency, and in this paper, each value is the annual runoff rate. The null hypothesis is that there is no change point, and the alternative hypothesis is that a change points exist. The test statistic of Pettitt's test is as follows,

$$K_N = \max_{1 \le j \le N} \left| U_{j,N} \right| \tag{1}$$

where

$$U_{j,N} = U_{j-1,N} + \sum_{k=1}^{N} sgn(x_j - x_k) \text{ for } j = 2, \cdots, N \tag{2}$$

and

$$sgn(x_j - x_k) = \begin{cases} 1 & if\ x_j > x_k \\ 0 & if\ x_j = x_k \\ -1 & if\ x_j < x_k \end{cases} \tag{3}$$

The associated probability (P) is derived as follows:

$$P \cong 2 \exp\left(\frac{-6K_N^2}{N^3 + N^2}\right). \tag{4}$$

Thus, the probability value ($p$-value) derived from the test statistic of Pettitt's test can be calculated. From this, trend differences in two parametric groups, $x_1, \cdots, x_j$ and $x_{j+1}, \cdots, x_N$, are analyzed, and the point associated with the test statistic ($K_N$) is the change point of the time series trend.

The Mann–Kendall test was used to determine whether there was a monotonic increase of decrease trends in the long-term time series data ($x_1, \cdots, x_n$). This method is used to detect a monotonic trend, where the null hypothesis is that there are no monotonic trends and the alternative hypothesis is that monotonic trends exist. The test statistic of the Mann–Kendall test is calculated according to,

$$S = \sum_{k=1}^{n-1} \sum_{j=k+1}^{n} sgn(x_j - x_k) \tag{5}$$

where $n$ is the amount of time series data. The Mann–Kendall statistic $S$ follows a normal distribution, and the following Z-transformation is employed,

$$Z = \begin{cases} \frac{S-1}{\sigma} & if\ S > 0 \\ 0 & if\ S = 0 \\ \frac{S-1}{\sigma} & if\ S < 0 \end{cases} \tag{6}$$

where the variance of $S$ is:

$$\sigma^2 = \frac{1}{18} n(n-1)(2n+5). \tag{7}$$

In other words, when a $z$-value exists within the rejection region, the null hypothesis is rejected, and the time series data show a monotonic trend.

*2.5. Baseflow Separation*

There has been much debate about whether forest thinning would increase flooding. In particular, for sustainable clean water supply, it is necessary to analyze the direction of changes caused by forest thinning. Therefore, this study conducted a more in-depth analysis of the changes of runoff components after forest thinning using baseflow separation analysis.

The streamflow at any time ($Q_t$) can be separated into quickflow ($A_t$) and baseflow ($B_t$) [26]:

$$Q_t = A_t + B_t. \tag{8}$$

There are several methods for baseflow separation analysis. The Eckhardt filter can separate baseflow from streamflow with a simple digital filter, and it describes forest catchment characteristics well [7,27]. The Eckhardt filter equation is as follows,

$$B_t = \frac{(1 - BFI_{max})aB_{t-1} + (1-a)BFI_{max}Q_t}{1 - aBFI_{max}} \tag{9}$$

where $B_t$ is the baseflow for time $t$ (mm), $Q_t$ is the streamflow for time $t$ (mm), and $a$ is the recession constant. $BFI_{max}$ is the maximum value of the baseflow index. In porous aquifers such as the GCP and GND forest catchments, a 0.8 $BFI_{max}$ value was suggested [27]. And the AR (1) model, which explains

the baseflow at one point from a previous one, for calculating the recession constant. The AR (1) model is appropriate for analyzing the baseflow characteristics in the small forest catchment [7].

## 3. Results

### 3.1. Changes of Annual Rainfall-Runoff Characteristics

　　　Average annual runoff rates for the GCP and GND forest catchments during the entire experimental period are about 45.6%, and 61.2%, respectively (Table 3). The relatively high annual runoff rates are due to the hydrological characteristics of the Republic of Korea, in which over 70% of the annual rainfall falls intensively in the summer monsoon rainy season (June to September), and the surface soil layers are relatively shallow. Furthermore, the steep slopes of streams cause floods to runoff quickly. Therefore, runoff rates are relatively high compared to other countries.

　　　Both GCP and GND forest catchments have significant variations in annual rainfall and runoff over time, and as the annual rainfall increases, the annual runoff also increases (Figure 2). In particular, with the exception of the last four years (2014~2017), when an unusual drought occurred in Korea, the annual rainfall in both GCP and GND forest catchments continuously increased depending on the effects of climate change. Because of this trend, the annual runoff also showed a gradual increase. The Mann–Kendall test was conducted to verify this statistically. Rainfall showed a monotonic increase tendency in GCP and GND forest catchments ($p < 0.01$), and for runoff and runoff rates, the GCP forest catchment showed no monotonic increase or decrease tendency ($p = 0.19$, 0.78 respectively), while the GND forest catchment showed a monotonic increment tendency during whole periods ($p < 0.01$, $p = 0.046$ respectively).

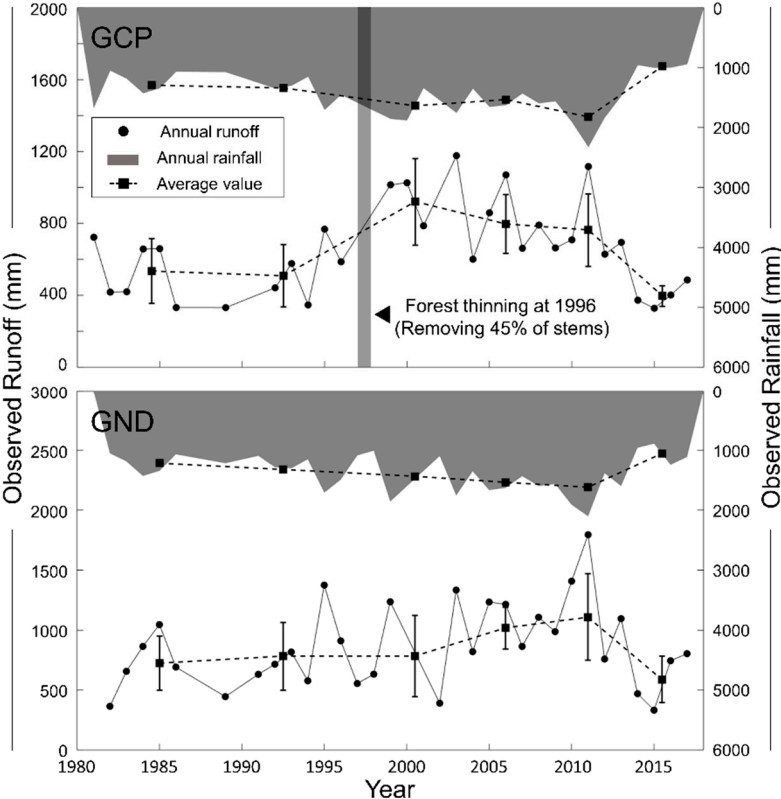

**Figure 2.**　Long-term annual rainfall-runoff in GCP and GND forest catchments. In the GCP forest catchment, runoff rapidly increased after 1996 when forest thinning was conducted. However, runoff form the GND forest catchment did not change very much. In both catchments, annual rainfall-runoff data, including missing values from natural disasters, were excluded from the graph.

**Table 3.** Rainfall-runoff characteristics in GCP and GND forest catchments.

| Periods * | GCP Forest Catchment | | | GND Forest Catchment | | |
|---|---|---|---|---|---|---|
| | Rainfall | Runoff | Runoff Rates | Rainfall | Runoff | Runoff Rates |
| 1981~1988 | 1293 | 535 | 41.4 | 1216 | 724 | 59.5 |
| 1989~1996 | 1340 | 509 | 38.0 | 1321 | 781 | 59.1 |
| 1997~2004 | 1638 | 921 | 56.2 | 1434 | 783 | 54.6 |
| 2005~2008 | 1533 | 797 | 52.0 | 1535 | 1018 | 66.3 |
| 2009~2013 | 1823 | 763 | 41.9 | 1617 | 1109 | 68.6 |
| 2014~2017 | 978 | 397 | 40.6 | 1051 | 587 | 55.9 |

* Periods were arbitrarily divided.

Comparing the differences between both catchments, the runoff rates of the GCP forest catchment were lower than that of the GND forest catchment, from 1981 to 1996, before the forest thinning was conducted (Table 3). However, since 1996, the runoff rates of the GCP forest catchment have increased, unlike the GND forest catchment, whose runoff rates are almost the same as before. For years, the runoff rates of the two catchments have been similar. In particular, the GCP forest catchment showed a sharp increment in rainfall, runoff and runoff rates after 1996; since then, the increment of runoff rates in the GCP forest catchment gradually decreased, and since 2009, the runoff rates have become similar to the GND forest catchment again. As a result, unlike the GND forest catchment, where the runoff rates are relatively uniform over time, the runoff rates of the GCP forest catchment significantly increased after the forest thinning.

*3.2. Runoff Characteristic Shifts and Change Point Determination*

3.2.1. Double Mass Curve of Rainfall-Runoff Measurements

The double mass curve method was used to analyze the rainfall-runoff characteristic changes in GCP and GND forest catchments (Figure 3). Because the slope of the double mass curve represents the runoff rates, the whole period from 1981 to 2017 was divided into three periods, and slopes of regression lines in each period were analyzed (Table 4).

After the forest thinning, in the GCP forest catchment, the slope of the double mass curve increased from 0.39 to 0.56, and the slope was nearly the same as that of the pre-thinning period, at 0.41 since 2009. In other words, there has been a drastic change in the slope since 1996 when the forest thinning took place, and the slope gradually decreased over time, making it similar to the period before forest thinning. From this curve, the runoff rates from 1997 to 2008 increased by 17%, compared to the period before 1996. On the other hand, in the GND forest catchment, slopes of the regression lines over time were 0.60, 0.66, and 0.65, respectively, with no sharp changes over the entire period. In other words, the GND forest catchment has seen gradual changes in its slope over the entire period.

**Table 4.** Linear relations between cumulative rainfall ($\sum P$) and runoff ($\sum R$) measurements in GCP and GND forest catchments ($\sum Q = a \sum P + b$).

| Periods | GCP Forest Catchment | | | GND Forest Catchment | | |
|---|---|---|---|---|---|---|
| | a | b | R2 | a | b | R2 |
| 1981~1996 | 0.39 | 115.85 | 0.9982 | 0.60 | −267.64 | 0.9974 |
| 1997~2008 | 0.56 | −2501.23 | 0.9985 | 0.66 | −1363.99 | 0.9983 |
| 2009~2017 | 0.41 | 1818.78 | 0.9988 | 0.65 | −532.19 | 0.9920 |

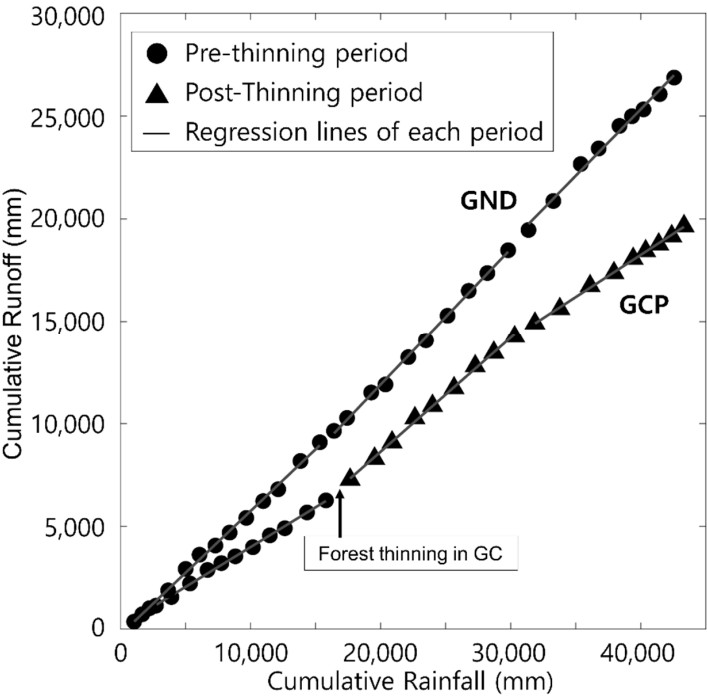

**Figure 3.** Double mass curves of GCP and GND forest catchments.

### 3.2.2. Forest Thinning Effect on Water Yield Increase

Pettitt's test was performed to determine the objective change point of the annual rainfall-runoff. Figure 4 shows the $U_{j,N}$ values in chronological order. The dashed lines represent the 10% significance level and the dotted lines represent the 5% significance level. All $U_{j,N}$ values in the GND forest catchment are not located in the 10% rejection region, which means that there are no significant changes in the annual rainfall-runoff characteristics over the entire period. Although, not all values of the GCP forest catchment were located within the rejection region of a 10% significance level, the test statistic was very close to the rejection line in 1996, when the forest thinning was conducted.

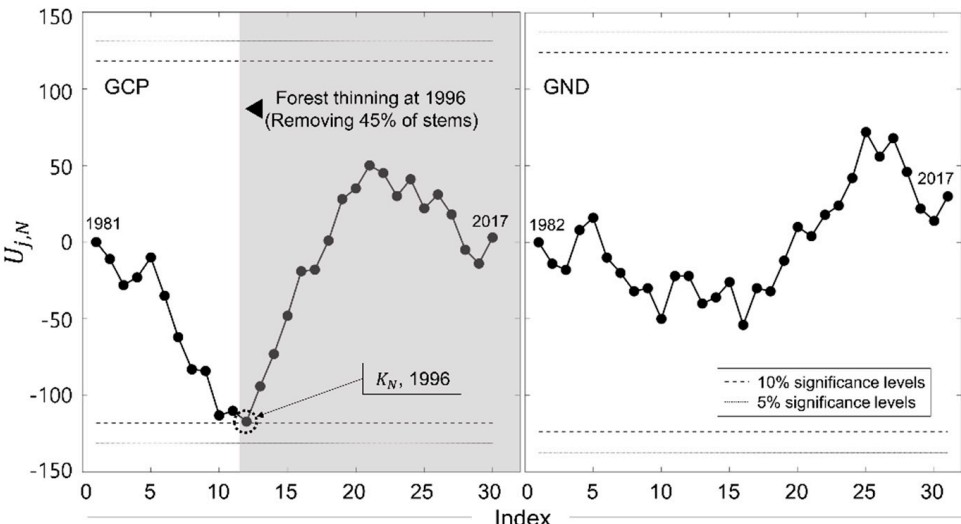

**Figure 4.** Pettitt's test of the GCP and GND forest catchments. The test statistics of the GCP forest catchment correspond to the value in 1996; the test statistic value does not fall in the rejection region ($p = 0.11$), but it is located near the 10% significance level. On the contrary, in the GND forest catchment, there is no significant change over the whole period.

Annual runoff in the GCP forest catchment can be seen to have a gradual decrease over time after the forest thinning was performed. Therefore, Pettitt's test was examined using data up to 2008, which showed a sharp increment in runoff characteristics (Figure 5). The test statistic corresponding to 1996 was located within the 1% significance rejection region, which means that the runoff characteristics from 1981 to 1996, and characteristics from 1997 to 2008, are statistically significantly different. Also, the negative value of the test statistic indicates that runoff rates have risen sharply since 1996.

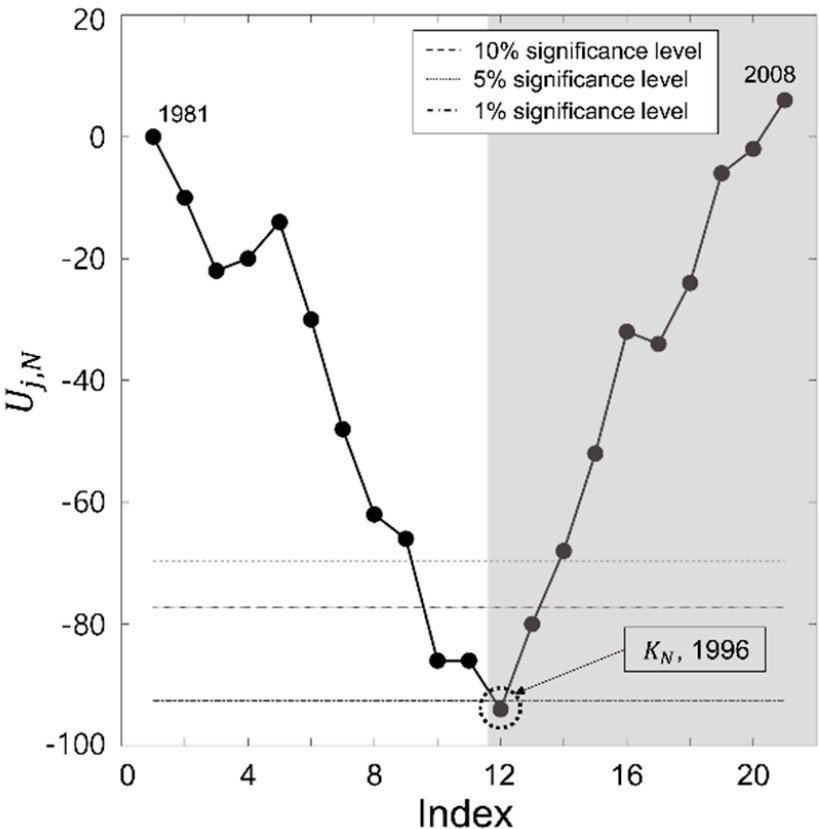

**Figure 5.** Pettitt's test of the GCP forest catchment based on annual rainfall-runoff data from 1981 to 2008. The test statistic corresponding to 1996 falls in the rejection region ($p = 0.008$), which means that runoff characteristics changed after 1996 and were statistically significant.

### 3.2.3. Long-Term Runoff Changes After Forest Thinning

From Figure 5, we can see a sharp increase in the runoff rates of the post-thinning period, compared to the period before the forest thinning in the GCP forest catchment. To see how the annual runoff characteristics changed over time after forest thinning was conducted, the Pettitt's test was examined based on the runoff rate data from 1997 to 2017 (Figure 6). In this case, the test statistic was located in the rejection region of the 10% significance level, and the change point corresponded to 2008. In other words, the runoff in the GCP forest catchment increased after 1996, due to forest thinning and runoff decreased since 2008, which was statistically significant.

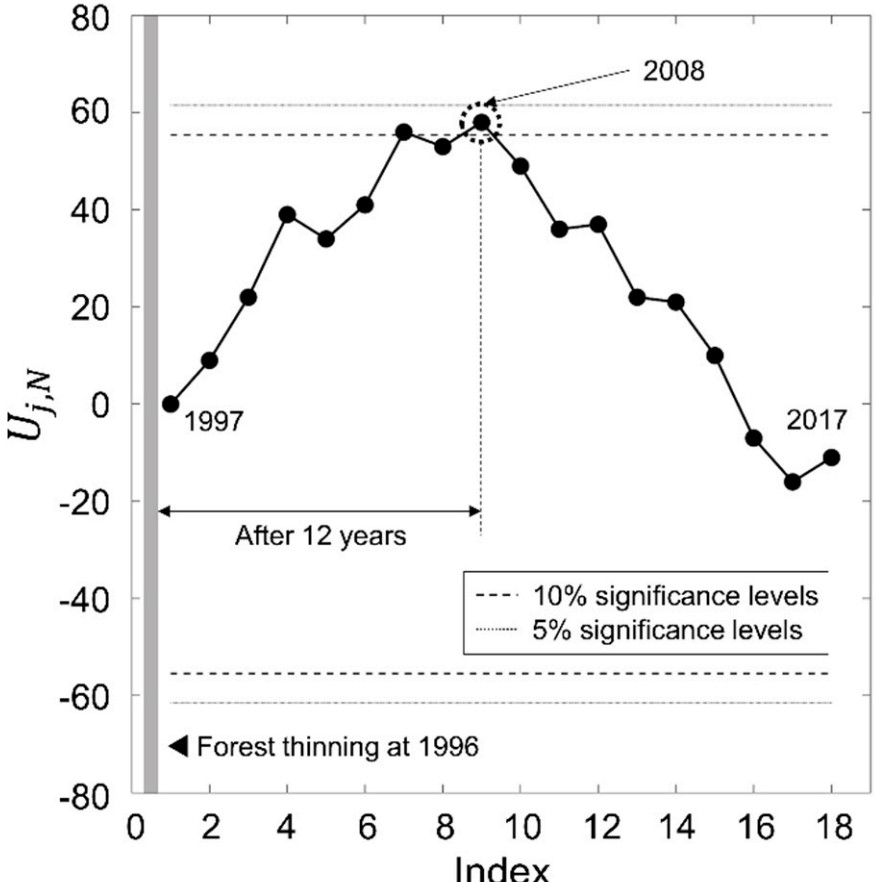

**Figure 6.** Pettitt's test of the GCP forest catchment after forest thinning. The test statistics corresponding to 2008 are in the rejection region.

### 3.3. Net Runoff Increment of Forest Thinning

The average annual runoff of the GCP forest catchment has increased by 366 mm for 12 years (1997 to 2008) compared with the pre-thinning period (1981 to 1996) due to the forest thinning conducted in 1996. In terms of this, it increased by 70%. Considering that the average annual precipitation in Korea is 1343 mm, this is a very large increment. In addition, compared with previous research, we can see that it is a very large water yield increment [10,15].

The Mann–Kendall test showed a gradual increase in annual rainfall changes in the GCP forest catchment. Thus, a large increment in water yield clearly exists due to the effect of increased rainfall due to the climate change, and an annual runoff increase of 366 mm should not be considered as the net runoff increment of forest thinning. The effect of rainfall change should be excluded in order to quantify the effect of the net increase. In other words, the impact of increased rainfall and the impact of forest thinning should be divided and assessed. For this purpose, we assumed that the factors which have the greatest impact on the increase of annual runoff in the GCP forest catchment are the precipitation increase and the forest thinning.

To exclude the effects of rainfall, the simple linear regression model from the double mass curve was used. The double mass curve already contains rainfall data on the x-axis, so the regression model based on it can exclude the effects of rainfall on the annual runoff changes [22]. Therefore, a simple linear regression model was created based on the data from 1981 to 1996 before the forest thinning was performed, and the actual measurements and model values were compared for each period. Based on the years 1996 and 2008, when the annual runoff characteristics were statistically changed, the entire period was divided into three parts to compare the observed and modelled values (Figure 7).

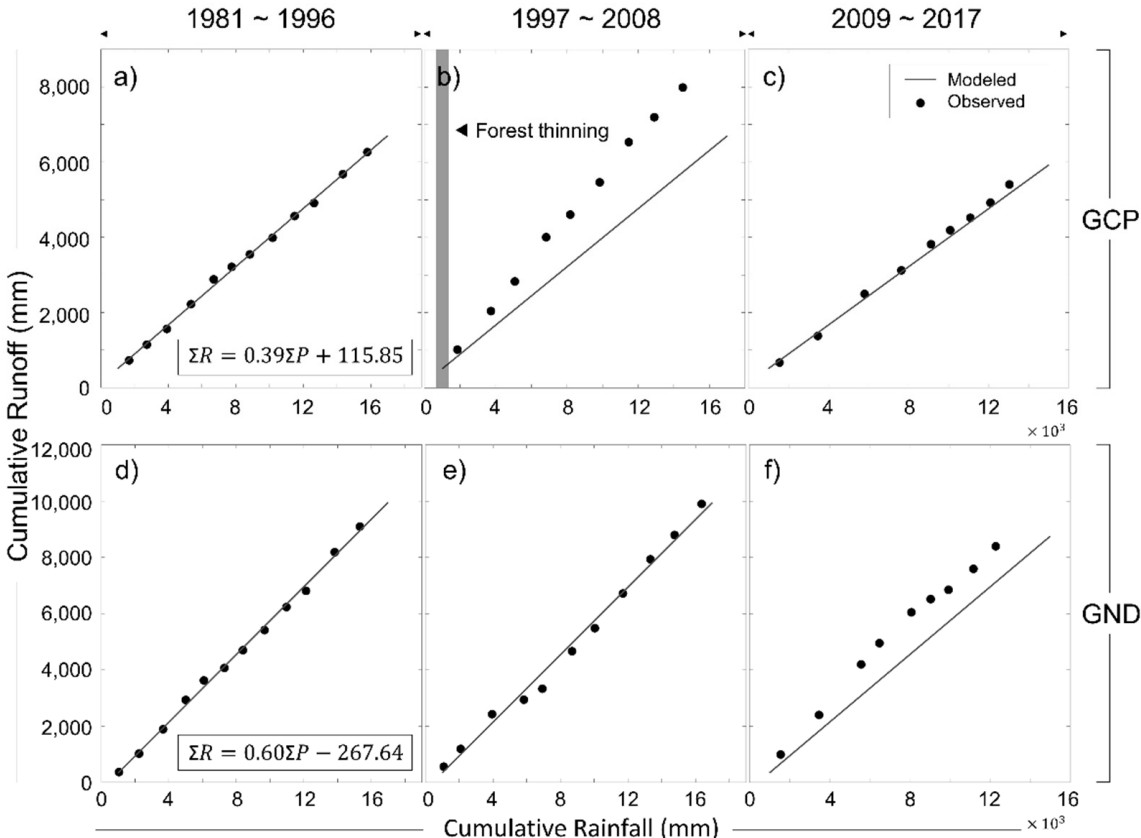

**Figure 7.** Simple linear regression model of double mass curve and observed values in the GCP (**a–c**) and the GND (**d–f**) forest catchments. In the second period in the GCP forest catchment (1997~2008) after forest thinning was conducted, there is a radical discrepancy between the observed and modelled runoff.

From 1981 to 1996, a simple linear regression model was created based on the annual rainfall-runoff double mass curve in the GCP forest catchment (Figure 7a; $\sum R = 0.39 \sum P + 115.85$). Based on this regression model, the comparison of the observed with modelled values for the remaining second period resulted in a sharp increase in the amount of runoff after the forest thinning was performed, but there was no significant difference between the observed and modelled values in the third period after a certain period of time after the forest thinning (Figure 7b,c); that is, based on Figure 7b, the net annual increase in runoff is 263 mm, accounting for about 72% of the total runoff increase of 366 mm (Table 5). The simple linear regression model of the GND forest catchment is $\sum R = 0.60 \sum P - 267.64$ (Figure 7d). Although the simple linear regression model accounts for many parts of the observed values, it can be seen that there is a difference between the modelled and observed values over time. In particular, the third period (2008 to 2017; Figure 7f) shows that the observed runoff is higher than the modelled runoff.

**Table 5.** Precipitation and thinning impacts on the water yield increase in the GCP forest catchment.

| Periods | Observed Runoff | Modelled Mean | Total Change | | Rainfall Impact | | Thinning Impact | |
|---|---|---|---|---|---|---|---|---|
| | | | Amount | Per * | Amount | Per * | Amount | Per * |
| 1981~1996 | 522 | – | – | – | – | – | – | – |
| 1997~2008 | 888 | 624 | 366 | 70.0 | 102 | 28.0 | 263 | 72.0 |
| 2009~2017 | 600 | 561 | 78 | 8.8 | 39 | 49.2 | 40 | 50.8 |

* percentage (%).

To confirm how the runoff characteristics of the GCP forest catchment changed over time, the cumulative difference between the observed and modelled runoff using the simple linear regression model was plotted with the cumulative rainfall (Figure 8). Until 1996, only minor errors existed, but the runoff has increased sharply since the forest thinning was carried out. Subsequently, the slope of the graph gradually decreases, and over time, the water yield increase effect gradually disappears.

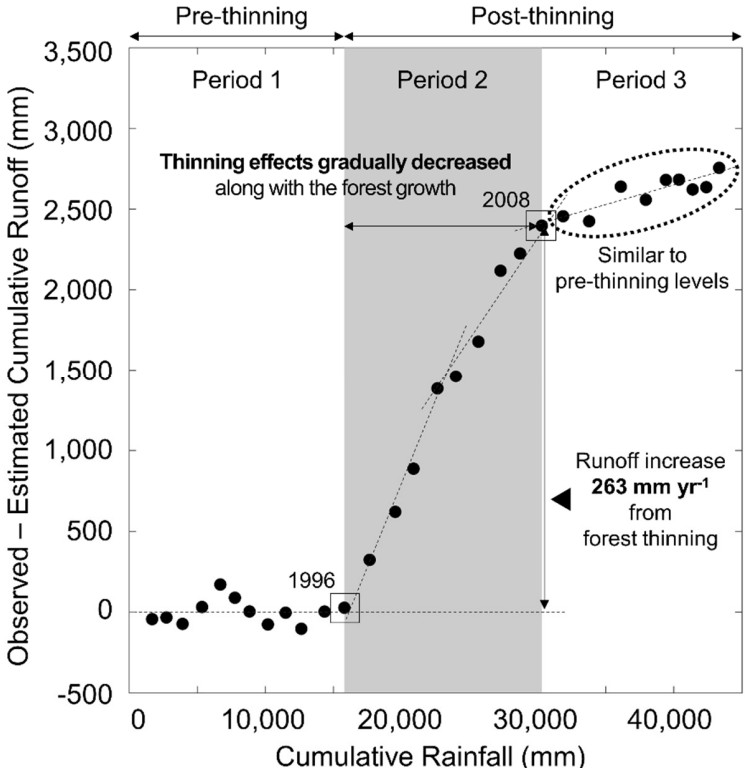

**Figure 8.** Relationship between cumulative runoff discrepancy and rainfall. Runoff increased after forest thinning, and the thinning effect gradually decreased after 12 years of runoff change.

### 3.4. Baseflow and Quickflow Changes of Each Period

Based on Section 3.2 to Section 3.3, a forest thinning effect on water yield changes was detected with statistical analysis, and the net water increment was quantified. To identify the forest thinning effect on the changes of baseflow and quickflow, baseflow separation analysis was conducted in the rainy season for each period, which is separated based on the change points in which the runoff had statistically significantly changed. Recession constants were calculated from recession curves with the AR (1) method, and baseflow was separated from the streamflow with the Eckhardt filter (see Equation (9)). The baseflow index (BFI) is the ratio of the amount of baseflow and streamflow and the quickflow ratio is the ratio of the amount of quickflow and streamflow. Because the streamflow can be divided into two components, baseflow and quickflow, the sum of the two ratios of the two components always amounts to 1 (see Equation (8)).

In the GCP forest catchment, BFI was 0.70 in the pre-thinning period (Table 6). However, the BFI increased by about 6% after forest thinning and again decreased by the same amount before forest thinning (0.69). In the same manner, the quickflow ratio was 0.30 in the first period and decreased by about 6% after the forest thinning; it was 0.31 in the third period, in which the quickflow ratio increased again. On the other hand, the BFI and quickflow ratio of the GND forest catchment did not change over time (BFI values were 0.72, 0.72, and 0.71, respectively).

**Table 6.** Rainfall-runoff characteristics in the rainy season of each period in the GCP and GND forest catchments.

| Rainfall-Runoff Characteristics | GCP | | | GND | | |
|---|---|---|---|---|---|---|
| | Period1 | Period2 | Period3 | Period1 | Period2 | Period3 |
| Year | 1981~1996 | 1997~2008 | 2009~2017 | 1981~1996 | 1997~2008 | 2009~2017 |
| Baseflow index (BFI) | 0.70 | 0.76 | 0.69 | 0.72 | 0.72 | 0.71 |
| Quickflow ratio | 0.30 | 0.24 | 0.31 | 0.28 | 0.28 | 0.29 |

Note: The baseflow index (BFI) is the rate of the amount of the baseflow to the streamflow, and the quickflow ratio is the rate of the amount of the quickflow to the streamflow; in Korea, the rainy season is June to September.

## 4. Discussion

### 4.1. Runoff Shifts in GCP and GND Forest Catchments

Comparing the runoff between the GCP and GND forest catchments, more runoff in the GND forest catchment was found than in the GCP forest catchment, when similar rainfall occurred. When the runoff and runoff rates during the pre-treatment period (before 1996) were compared between the two catchments, the annual rainfall characteristics were similar. However, the GND forest catchment produced 230 mm more water per year than the GCP forest catchment, and considering that the average annual rainfall in Korea is 1340 mm, the water supply in the GND forest catchment was 17% higher than the GCP forest catchment. The main difference between the two catchments is forest cover. The GCP forest catchment is a coniferous forest plantation. Whereas, the GND forest catchment is a natural deciduous forest. Because of this, the two catchments yield different amounts of water. Studying water supply changes according to forest type began with Swank [28]. In this study, coniferous trees were planted after the clear cutting of the deciduous catchment, which resulted in a decrease in water yield. In other words, although there is a deviation in terms of the forest characteristics such as tree density, leaf area index, forest age, etc., this generally means that the water yield of coniferous forests is less than that of deciduous forest, and related research has been steadily conducted to date [7,13,29].

In addition to the difference in the annual runoff characteristics of the two catchments, chronological changes in the annual rainfall-runoff characteristics were observed in each catchment. In the GCP forest catchment, the slope of the double mass curve became steeper and the annual runoff was increased after the forest thinning. However, it was not clear whether the change point of the runoff rates was in 1996 when the forest thinning was carried out. Pettitt's test confirmed statistically that the change point was 1996. By confirming that the year corresponding to the test statistic and the year in which the forest thinning was carried out are the same, we can determine that the reason for the increase in the runoff rates was the forest thinning in 1996.

From the baseflow separation analysis, we find that the quickflow ratio was reduced and baseflow ratio (BFI) was increased after the forest thinning in the GCP forest catchment. On the contrary, baseflow and quickflow ratios in the GND catchment did not change over time. In other words, after forest thinning, the total amount of streamflow increased, and the baseflow and quickflow also increased. However, based on from the decrease in quickflow ratio, we can confirm that a huge portion of the increased streamflow originated from the baseflow. Dung [10] also confirmed that the runoff increase from forest thinning was highly associated with the baseflow component.

Conversely, in the GND forest catchment conserved naturally, the Mann–Kendall test showed that annual rainfall, runoff, and runoff rates gradually increased over time. The increasing trend in annual rainfall can be explained by climate change. Changes in rainfall characteristics, caused by climate change, have been mentioned in many studies [30], and the increase in annual rainfall has been observed in many regions around the world [31–33]. The increase in runoff can be explained by the reduction of the water consumption of the old, matured forest, rather than the impact of climate change [34–36].

### 4.2. Decreasing Forest Thinning Effect with Forest Growth

Tree growing is a major factor of the decreasing forest thinning effect. During the period forest thinning in 1996, it was observed that where the mean tree density decreased, the mean tree height, mean DBH, and mean stem volume slightly increased. However, the growing stock volume and the crown closure greatly decreased. Until 2003, however, the crown closure drastically increased to 95.0%, and it is at almost the same level as the pre-thinning period. After that, the crown closure remained at similar levels (96.0% in 2010 and 95.0% in 2016). The growing stock volume also greatly increased after forest thinning. Almost 66% of the growing stock volume increased from 1996 to 2003, and the growth rates subsequently slowed down. The growth in trees increases the total amount of evapotranspiration. From the water balance mentioned above, increasing evapotranspiration makes forest runoff decrease. Consequently, the introduction of understory plants and growth of individual trees, as forest growth, result in increasing evapotranspiration and decreasing forest runoff.

### 4.3. Quantifying the Net Runoff Increment in the GCP Forest Catchment

The double mass curves and simple linear regression model successfully excluded the effect of rainfall changes from total runoff changes. From this model, it can be interpreted that, in the GCP forest catchment, 28% of the effects of rainfall increase and 72% of effects of forest thinning result in a 366 mm total water yield increase. Thus, the effect of forest thinning carried out in 1996 is greater than the rainfall increase due to climate change, and the net runoff increment from the forest thinning is 263 mm yr$^{-1}$. Appropriate forest management is important for the sustainable supply of fresh water resources in response to the effects of climate change [32]. While, climate change is a natural phenomenon that humans cannot control, we can decide how to manage forests with proper direction. From this point of view, the results of the study showed that in a coniferous forest plantation in Korea, forest thinning can successfully increase water yield and can have a greater impact than the effects of annual rainfall changes derived by climate change.

However, rainfall change and the forest thinning are not the only reasons for the changes in annual runoff characteristics. Changes in forest stand age, including those outlined in studies by Kuczera [34] and the results of the GND forest catchment, can also affect runoff characteristics. There may be more factors affecting the annual water yield that have yet to be identified. Since the above results were derived by assuming that the most important factors affecting the annual runoff changes are the rainfall and forest thinning, a limitation in this paper is the failure to take into account other factors in calculating the net runoff increment. Thus, additional studies are required for analyzing the net water yield increment.

## 5. Conclusions

The effects of forest thinning on long-term runoff was confirmed based on the long-term rainfall-runoff data in the GCP and GND forest catchments in Korea. The double mass curve (DMC) and Pettitt's test showed that, in the GCP forest catchment, the slope of the double mass curve changed in 1996 and 2008, proving statistically significant. Forest thinning increased the annual runoff rates, and the effect of forest thinning gradually decreased, which lasted for about 12 years. This was because the forests have grown rapidly after forest thinning and the canopy closed quickly. A simple linear regression model of the double mass curve can successfully quantify the net effect of the forest thinning. As a result, the total runoff increase is derived from 72% of the forest thinning impacts and 28% of the rainfall increase. The net runoff increment effect was 263 mm yr$^{-1}$ and, from the baseflow separation analysis in which the quickflow ratio decreased after forest thinning. It can be confirmed that forest thinning does not significantly increase the amount of the quickflow. On the contrary, the GND forest catchment showed no significant changes in the runoff characteristics in the DMC and the Pettitt's test. In this study, the water yield increment by forest thinning was statistically identified and considered to be more influential than the increased rainfall caused by climate change. This could suggest the

direction that forest policy for water resource management is being undertaken in the Republic of Korea's future. However, the interaction between plant growth and the increase in rainfall could not be considered in this study, and only the effect of the rainfall change was excluded in quantifying the net runoff increment. Thus, additional studies should be conducted in future research.

**Author Contributions:** Conceptualization, H.Y. and H.T.C.; methodology, H.Y.; software, H.Y.; validation, H.Y. and H.T.C.; formal analysis, H.Y.; investigation, H.Y., H.T.C. and H.L.; resources, H.T.C. and H.L.; data curation, H.T.C. and H.L.; writing—original draft preparation, H.Y.; writing—review and Editing, H.Y. and H.T.C.; visualization, H.Y.; supervision, H.T.C.; project administration, H.T.C. and H.L.; funding acquisition, H.T.C.

**Funding:** This research received no external funding.

**Conflicts of Interest:** The authors declare no conflict of interest.

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
