# Peer review of "Effects of Forest Thinning on the Long-Term Runoff Changes of Coniferous Forest Plantation"

_water, doi:10.3390/w11112301_

Round 1
Reviewer 1 Report
The paper is focused on the topical issue of the hydric function of forests and the possibilities of their influence by silvicultural measures such as thinning. From this perspective, the topic is suitable for publication in a scientific journal.
However, the hydrological function of the forest is much wider than just the effect of the forest on water runoff. Therefore, I recommend the authors to extend the introduction of the article with a more detailed definition of this issue in its entirety.
The methodology should describe in more detail how the water balance of the stands was measured and what runoff this actually is. This cannot be reliably inferred from what is stated here.
In addition, the description of the stands and the intervention should be more detailed. The methodology lacks data on basic parameters of stands, such as mean height, diameter, volume and stock. Only the number of trees removed describes thinning, but there is no indication of what type of selection was used and how much biomass was removed by the intervention. The size of crowns, resp. the amount of assimilation apparatus are very important for interception, these data are also missing.
In describing the statistical methods that were used, there is probably an error in Equation 3, where the last line should be an expression:
If xj < xk
The results are interesting; in general, a very high share of runoff in total rainfall is surprising (in the case of farm forest mostly above 40%, in natural forest above 55%). However, for the assessment of this magnitude, there is no detailed characteristics of the outflow (what is included in it). The size of the runoff is also strongly influenced by the intensity of individual rainfall (precipitation), so this information would be helpful to understand the results.
The authors have positively interpreted the significant impact of thinning on the increase in runoff. However, it is necessary to realize that one of the important aspects of the forest's hydrological function is, on the other hand, the damping of flood waves, which is possible only by reducing the surface runoff. From this perspective, it would be useful to separate the individual runoff kinds and quantify them separately.
For the reasons set out above, I propose to revise the manuscript according to the above comments.
Author Response
Dear reviewer 1
Thank you for reading this paper carefully and providing profound advice.
We thought deeply about your comments and tried hard to improve this manuscript.
From the abstract to the conclusions, we have revised many parts of the previous manuscript based on your advice.
We also used the English Editing Service, operated by the MDPI (English Editing ID: English-13223), to provide our explanation clearly to readers.
Regarding the information you have mentioned, I responded to your suggestions and comments in the attached file below.

Reviewer 2 Report
The whole manuscript is quite acceptable. I only have several suggestions:
Move the figures and the related expression in the Discussion part to the Result part, and reorganize them accordingly. Shorten the Conclusion part, and not only include the achievements but also the limitations of the current study (such as didn't consider the interaction between plant growth and the increase of rainfall). Delete sentences such as "The annual rainfall and runoff data of the Gwangneung coniferous plantation (GCP) and Gwangneung natural deciduous (GND) forest catchments collected since 1981 are presented in Figure 2." Rewrite the sentence: Before the forest thinning, 2,102 trees were 112 planted in 1 ha but 1,147 were left after the forest thinning. Lines 256-257, "GND produced 230 mm more water per year than"? Lines 302-303, sentence like "we assumed that the factors" should be placed earlier, such as in the Introduction part. Lines 339-340, sentence like "Forest thinning is one of these forest managements and..." has been mentioned several times before, so, please avoid in there.
Author Response
Dear reviewer 2
Thank you very much for your positive feedback and for providing profound advice.
We also thank you for reading this paper carefully and suggesting to me how to better explain this paper to readers.
And we used the English Editing Service, operated by the MDPI (English Editing ID: English-13223), based on your comments related with English expressions.
Having the profound thinking you have mentioned, I responded to your suggestions and comments in the attached file below.

Round 2
Reviewer 1 Report
New version of the manuscript accepted almost all my requirements and comments.
So, I have only two comments:
In the chapter 4.2. Decreasing Forest Thinning Effect with Forest Growth this statement is written:
This growth in trees increases the total amount of evapotranspiration. From the water balance mentioned above, increasing evapotranspiration makes forest runoff increase. My question is: It is really true or is it the other way around?
In the Conclusion (chapter 5) this statement is written:
The net runoff increment effect was 263 mm.yr-1 and, from the baseflow separation analysis in which the quickflow ratio was decreased after the forest thinning, it can be confirmed that the forest thinning does not significantly increase the risk of flooding.
My question is:
Have you any explanation of this observation? By my opinion after thinning (which reduce crown canopy), rather increase of the quickflow could be expected.
Please, take into your consideration these questions.
Author Response
Dear reviewer 1
Thank you for your important advice and comments.
We thought deeply about your questions and revised our manuscript to fully reflect your comments.
We modified some parts of the discussion and conclusion and added a brief description.
I responded to your questions and comments in the attached file below.
